# Genetically Modified Ferritin Nanoparticles with Bone-Targeting Peptides for Bone Imaging

**DOI:** 10.3390/ijms22094854

**Published:** 2021-05-03

**Authors:** Jong-Won Kim, Kyung-Kwan Lee, Kyoung-Woo Park, Moonil Kim, Chang-Soo Lee

**Affiliations:** 1Bionanotechnology Research Center, Korea Research Institute of Bioscience & Biotechnology (KRIBB), Daejeon 34141, Korea; kimjw@kribb.re.kr (J.-W.K.); lkk@kribb.re.kr (K.-K.L.); qkrruddn94@kribb.re.kr (K.-W.P.); 2Department of Life and Nanopharmaceutical Science, College of Pharmacy, Kyung Hee University, Seoul 02447, Korea; 3Department of Biotechnology, University of Science & Technology (UST), Daejeon 34113, Korea

**Keywords:** bone-targeting peptides, ferritin nanoparticles, osteoblasts, hydroxyapatite, bioimaging

## Abstract

Bone homeostasis plays a major role in supporting and protecting various organs as well as a body structure by maintaining the balance of activities of the osteoblasts and osteoclasts. Unbalanced differentiation and functions of these cells result in various skeletal diseases, such as osteoporosis, osteopetrosis, and Paget’s disease. Although various synthetic nanomaterials have been developed for bone imaging and therapy through the chemical conjugation, they are associated with serious drawbacks, including heterogeneity and random orientation, in turn resulting in low efficiency. Here, we report the synthesis of bone-targeting ferritin nanoparticles for bone imaging. Ferritin, which is a globular protein composed of 24 subunits, was employed as a carrier molecule. Bone-targeting peptides that have been reported to specifically bind to osteoblast and hydroxyapatite were genetically fused to the N-terminus of the heavy subunit of human ferritin in such a way that the peptides faced outwards. Ferritin nanoparticles with fused bone-targeting peptides were also conjugated with fluorescent dyes to assess their binding ability using osteoblast imaging and a hydroxyapatite binding assay; the results showed their specific binding with osteoblasts and hydroxyapatite. Using in vivo analysis, a specific fluorescent signal from the lower limb was observed, demonstrating a highly selective affinity of the modified nanoparticles for the bone tissue. These promising results indicate a specific binding ability of the nanoscale targeting system to the bone tissue, which might potentially be used for bone disease therapy in future clinical applications.

## 1. Introduction

Bones, a rigid tissue that make up the body skeleton, play a major role in supporting and protecting various organs as well as a body structure. The bone density is consistently regulated by balancing the functions of bone-forming osteoblasts and bone-resorbing osteoclasts, which is also known as bone remodeling [1,2]. However, if the balance of differentiation and functions between osteoblasts and osteoclasts collapses, it can lead to several skeletal diseases such as osteoporosis, rheumatoid arthritis, and bone metastases [3,4,5,6,7,8,9]. Recently, osteoblasts, osteoclasts, and hydroxyapatite have been employed as targets for bone-specific drug delivery for the inhibition of bone resorption and stimulation of bone formation. Anti-resorption agents, such as bisphosphonates including alendronate as a calcium targeting ligand, have been widely used for the treatment of osteoporosis, preventing osteoporotic fracture [10]. To achieve effective diagnosis and therapeutics for bone diseases, osteoblast stimulation is deemed necessary; improved and efficient osteoblast-targeted delivery systems are required to accomplish this task. Therefore, osteoblasts and hydroxyapatite have been considered as important targets.

Various kinds of nanostructures, such as polymer nanoparticles, liposomes, and inorganic or carbon-based nanoparticles, have been studied for their usability in drug delivery systems and imaging, and as biosensors [11,12,13,14,15]. Specifically, synthetic nanoparticles are usually modified via chemical conjugation or surface treatment to achieve biological functions [16,17]. However, these nanoparticles show substantially low biocompatibility and high cytotoxicity, which are not suitable for in vivo applications. In addition, the chemical modifications on the target are often associated with a low delivery efficiency due to random orientation and instability of the modified molecules. Thus, the development of stable and controllable nanoparticles is required for an improved targeting ability.

Ferritin nanoparticles are hollow cage-like protein structures that naturally exist in the human body and are composed of nontoxic components [18]. They have recently drawn considerable attention due to their unique architecture and surface properties. A number of studies have been performed to analyze their usability based on internal cavity or external surface in the biomedical fields via either genetic or chemical modification; several important bioengineering applications have been reported thus far [19,20,21,22,23]. Thus, this approach is attractive in that the outer surface of ferritin nanoparticles could be chemically or genetically modified with targeting motifs, making the process of targeting more precise. Since the ferritin nanoparticles are composed of 24 subunits and can be genetically modified to expose multiple ligands onto the exterior surface, an eventual improvement in the binding affinity can be expected [24].

We herein present bone, especially osteoblast (OB-Fn) and hydroxyapatite (HA-Fn) targeting ferritin nanoparticles, synthesized via the genetic engineering of specific binding peptides. Peptides are promising targeting ligands due to their plausible low immunogenicity, high stability, simple bioconjugation, and ease of synthesis [25,26,27]. In this study, we have used these modified ferritin nanoparticles for bone-targeted imaging. The sequences of osteoblast and hydroxyapatite binding peptides, which are employed as targeting moieties, have been reported to be Ser-Asp-Ser-Ser-Asp (SDSSD) and six Asp residues ((Asp)6), respectively, in previous studies [28,29]. These bone-targeting peptides were genetically linked to subunits of ferritin nanoparticles and the resulting self-assembled ferritin nanoparticles formed a cage-like structure. Next, we demonstrated the specific binding of ferritin nanoparticles to osteoblasts and hydroxyapatite using cell imaging, calcium deficient hydroxyapatite (CDHA) scaffold binding assays, and ex vivo analysis. This work will serve to advance the use of ferritin nanoparticles as agents for drug delivery, bioassay, disease diagnosis, therapy, and molecular imaging with a particular focus on their bone-related biomedical applications.

## 2. Results and Discussion

### 2.1. Preparation and Characterization of Bone-Targeting Ferritin Nanoparticles

To achieve specific targeting for the bone tissue, SDSSD and (Asp)6, which are osteoblast and hydroxyapatite-specific binding peptides, respectively, were employed as the targeting moieties. A five-amino acid motif, SDSSD (Ser-Asp-Ser-Ser-Asp), was identified as osteoblast-specific binding peptide through the phage display method in a previous study [28]. Furthermore, acidic amino acids, such as aspartic acid and glutamic acid, are known for their interaction with hydroxyapatite by chelating calcium [29]. We selected a six aspartic acid-long peptide (Asp)6, due to its high binding affinity to hydroxyapatite. To prepare bone-targeting ferritin nanoparticles, each kind of the peptide was genetically fused to the N-terminus of the heavy chain ferritin subunit through a glycine-serine (GS) linker for their outward exposure on ferritin nanoparticles. As shown in Appendix A, the resulting ferritin nanoparticles were assessed using sodium dodecyl sulphate–polyacrylamide gel electrophoresis (SDS-PAGE). The bone-targeting ferritin nanoparticles were expressed as soluble proteins, reaching the expression level of around 30 mg per 1 L culture. Furthermore, a successful genetic fusion of the bone-targeting peptides to the ferritin subunit was indicated by an increase in the size of the subunit. We envisage that the insertion of five or six amino acids at the N-terminus of the ferritin subunit has no significant effect on the expression of ferritin nanoparticles. We subsequently investigated the morphology and size distribution of ferritin nanoparticles using transmission electron microscopy (TEM) and dynamic light scattering (DLS). As shown in Figure 1a,b, most peptide-conjugated ferritin nanoparticles typically exhibited spherical and cage-like structures, showing a uniformity in size and morphology. In addition, while the hydrodynamic size of WT-Fn was estimated to be approximately 11.7 nm, that of peptide-conjugated ferritin nanoparticles (OB-Fn and HA-Fn) was estimated to be approximately 13.6 nm, confirming the increase in the size of the nanoparticles due to peptides fused at the N-terminus (Figure 1c). These results indicated that bone-targeting ferritin nanoparticles were successfully constructed with a negligible effect of genetic fusion on the self-assembly of 24 subunits of ferritin.

### 2.2. In Vitro Imaging of Osteoblasts

Next, the selective binding ability of peptide-conjugated ferritin nanoparticles to osteoblasts was investigated using cell imaging. The osteoblast-targeting peptide has been known to bind periostin located on the membrane of the osteoblasts as described earlier [28]. Periostin, also known as osteoblast-specific factor 2 (OSF-2), plays a role in cell adhesion and migration in the bone tissue. It also stimulates osteoblastic differentiation, thereby leading to bone formation [30]. To carry out fluorescent imaging of osteoblasts, OB-Fn were conjugated with Cy5 at the cysteine residue inserted at the N-terminus of targeting peptides for efficient site-specific labeling. MC3T3-E1 cells, an osteoblast precursor cell line, were differentiated using application of ascorbic acid and β-glycerophosphate for 3 weeks. Undifferentiated MC3T3-E1 and HeLa cells were used as negative controls. As shown in Figure 2, upon incubation with OB-Fn, a strong fluorescent signal was observed only in the differentiated MC3T3-E1 cells, whereas a negligible fluorescent signal was detected in undifferentiated MC3T3-E1 and HeLa cells. In addition, no fluorescent signal was detected in the cells treated with Cy5-labeled WT-Fn due to absence of a targeting ability. It is known that osteoblastic differentiation regulates the level of periostin expression, which could affect the targeting efficiency of OB-Fn. Thus, these observations demonstrated the specific binding of the genetically modified ferritin nanoparticles, in which targeting peptides were successfully exposed outwards, to differentiated osteoblasts.

### 2.3. Binding Ability of Ferritin Nanoparticles to Hydroxyapatite

The binding affinity of ferritin nanoparticles to the hydroxyapatite surface was investigated using hydroxyapatite resins. Notably, natural amino acids, such as aspartic acid (Asp) or glutamic acid (Glu), are known to have a significant affinity for calcium phosphate of hydroxyapatite and repetitive aspartate sequences have been widely used as targeting moieties [31]. Based on this evidence, we incorporated a peptide with six aspartates, (Asp)6, into ferritin nanoparticles for specific binding to the hydroxyapatite. HA-Fn labeled with Cy5 were incubated with the hydroxyapatite resin followed by evaluation of a fluorescent signal. As a result in Figure 3, the hydroxyapatite treated with HA-Fn showed strong fluorescent signals as compared to that treated with WT-Fn. Approximately 10% of 12.5 μg/mL Cy5-labeled WT-Fn bound to hydroxyapatite, whereas approximately 72% of 12.5 μg/mL Cy5-labeled HA-Fn were estimated to bind to hydroxyapatite. For further investigation of the binding ability of the nanoparticles to hydroxyapatite, we performed a binding assay using 3D CDHA scaffolds fabricated from α-tricalcium phosphate (α-TCP) paste. Schematic diagram and method of CDHA scaffold fabrication is shown in Appendix A. α-TCP, owing to its high solubility and biodegradability, is being actively studied as a raw material for bone repair [32]. The binding ability of HA-Fn onto CDHA scaffolds was evaluated using a fluorescent microscopy; we observed an intense fluorescent signal from HA-Fn-treated CDHA scaffolds (Figure 4). In contrast, both WT-Fn and OB-Fn-treated scaffolds showed negligible fluorescent signals. These results indicated that HA-Fn, with outward exposed targeting peptides, could be effectively used as a bone-targeting material.

### 2.4. In Vivo Bone-Specific Targeting of Ferritin Nanoparticles

To verify the binding ability of ferritin nanoparticles to bone, Cy5-labeled OB-Fn and HA-Fn were intravenously administrated to the mice (10 mg/kg); Cy5-labeled WT-Fn were used as a negative control. After 24 h, all the mice were sacrificed and the major organs were collected from each mouse for detection of the fluorescence signal ex vivo. The fluorescent signals were clearly observed in the lower limbs of the mice injected with OB-Fn and HA-Fn, while the control mice treated with PBS and WT-Fn exhibited no fluorescent signals (Figure 5a). As shown in Figure 5b, the biodistribution of ferritin nanoparticles was also clearly identified and no significant accumulation in other organs, such as heart, lung, liver, spleen, and kidney, was apparent. These results supported previous observations that OB-Fn and HA-Fn were selectively able to bind to osteoblasts on hydroxyapatite and hydroxyapatite matrices, respectively. Thus, our genetically engineered Ob-Fn and HA-Fn can potentially be used for bone-targeted imaging or in vivo therapy. In the present study, we have demonstrated that ferritin nanoparticles were capable of targeting bone in addition to their tumor-targeted binding [33,34]. Compared to chemicals including bisphosphonate, these peptide conjugated nanoparticles might be advantageous in terms of reduced adverse effects including inflammation or pain in some organs during the treatment of bone-related diseases. Bone-specific targeting of peptide-conjugated ferritin nanoparticles can be easily extended by using various proteins through their genetic fusion at the terminus. Some growth factors or hormones, such as parathyroid hormones or transforming growth factor β (TGF- β), have shown outstanding effects on the treatment of osteoporosis by stimulating osteoblastic activities [35,36]. In addition, for diagnosis and monitoring the progress of osteoporosis, a functional near-infrared fluorescent probe has been developed to detect alkaline phosphatase (ALP), which is known as an osteoblast activity marker [37]. Diagnostic chemical probes can be attached to specific sites on ferritin nanoparticles through chemical conjugation. In some cases, functionalization of ferritin nanoparticles via genetic fusion might lead to formation of insoluble aggregates during expression in E. coli cells and require additional self-assembly steps [38]. Nevertheless, ferritin nanoparticles are fascinating materials due to their robust chemical and structural stability [39,40]. Furthermore, these nanoparticles with multivalent targeting moieties could improve the binding affinity to targets, resulting in highly sensitive and specific delivery [24]. Taken together, based on our results, ferritin nanoparticles have a considerable potential for further diagnostic and drug-delivery-related applications, specifically in the context of bone-related diseases.

## 3. Materials and Methods

### 3.1. Gene Expression and Protein Purification

Osteoblast and hydroxyapatite targeting ferritin nanoparticles (OB-Fn and HA-Fn) with 6X-His tag at the N-terminus, were cloned into a pET-21a vector (Invitrogen, Carlsbad, CA, USA). Osteoblast and hydroxyapatite binding peptides were genetically fused to the N-terminus of heavy chain ferritin subunits using a (GSS)_4_ linker. The constructed vector was transformed into a *Escherichia coli* host strain (BL21 (DE3)). The transformed cells were cultured in LB medium (BD Difco) containing 100 µg/mL ampicillin at 37 °C overnight and then diluted 1000-fold using fresh LB medium. The diluted cells were grown at 37 °C until the optical density at 600 nm reached 0.5 and then were induced by addition of IPTG (isopropyl β-D-1-thiogalactopyranoside) at a final concentration (0.5 mM) for protein expression. Cells were further incubated at 18 °C for 20 h and harvested by centrifugation at 4000 rpm. The collected cells were resuspended in a lysis buffer (20 mM Tris, 50 mM NaCl, 10 mM imidazole, pH 8.0) and disrupted using ultra-sonication. After centrifugation of cell lysate at 16,000 rpm and 4 °C for 50 min, the supernatant was collected and filtered through 0.2 µm syringe filter. The filtered supernatant was purified by affinity chromatography using a Ni-NTA resin. The solution was applied to the Ni-NTA column, followed by washing with a buffer (20 mM Tris, 50 mM NaCl, 40 mM imidazole, pH 8.0). The ferritin nanoparticles were eluted using an elution buffer (20 mM Tris, 50 mM NaCl, 250 mM imidazole, pH 8.0). The protein concentration was determined by measuring the absorbance of the solution at 280 nm and the proteins were stored at 4 °C till further analysis.

### 3.2. Transmission Electron Microscopy (TEM)

The morphologies of OB-Fn and HA-Fn were examined by field-emission transmission electron microscopy (FE-TEM) using a Tecnai G2 F30 S-TWIN microscope (FEI Company, Hillsboro, OR, USA) with an accelerating voltage of 200 kV. OB-Fn and HA-Fn were stained with phosphotungstic acid for TEM measurements. Briefly, the samples were prepared by dropping the solution containing OB-Fn and HA-Fn on form a var/carbon grid and subsequently evaporating it under ambient conditions. Phosphotungstic acid (2%, pH 7.4) was then dropped on the grid, the samples were stained for 1 min, and the acid was completely removed using a filter paper. After drying the grid further under ambient conditions, the images of OB-Fn and HA-Fn were obtained using a TEM microscope.

### 3.3. Measurement of Hydrodynamic Nanoparticles Size

OB-Fn and HA-Fn were diluted in 1× PBS buffer solution (pH 7.4) and subjected to dynamic light scattering (DLS) for determining the hydrodynamic size using the Zetasizer Nano ZS (Malvern, Worcestershire, UK).

### 3.4. Cell Culture

MC3T3-E1 (mouse pre-osteoblast) cells were purchased from ATCC (CRL-2593) and cultured in minimum essential medium alpha (α-MEM, Gibco, Grand Island, NY, USA) and HeLa (human cervical cancer) cells, which were obtained from Korean Collection for Type Cultures (KCTC) (HC18802), were grown in Dulbecco’s modified Eagle’s medium (DMEM, Gibco). Cells were cultured in medium containing 10% fetal bovine serum (GE Healthcare Hyclone, Logan, UT, USA) and 1% streptomycin/penicillin in 5% CO_2_ (MCO-5AC, Sanyo, Osaka, Japan) at 37 °C. For differentiation of the pre-osteoblasts, MC3T3 E1 cells were maintained in differentiation medium supplemented with 50 µg/mL ascorbic acid and 10 mM β-glycerophosphate by changing the medium every 2 or 3 days.

### 3.5. Fluorescent Dyes Labeling of Ferritin Nanoparticles

OB-Fn and HA-Fn, along with wild-type ferritin nanoparticles (WT-Fn) with no conjugated peptides as a negative control, were labeled with Cy5 for fluorescent imaging. Briefly, OB-Fn, HA-Fn, and WT-Fn were incubated with 30-fold molar excess of Cy5-maleimide in a buffer (20 mM Tris, 50 mM NaCl, pH 7.5) for 3 h at room temperature. Following the reaction, remaining free Cy5-maleimide was removed using a desalting column (PD-10). The resulting ferritin nanoparticles were concentrated using a centrifugal filter and stored at 4 °C for cell imaging and hydroxyapatite binding assay.

### 3.6. Fabrication of Calcium-Deficient Hydroxyapatite (CDHA) Scaffolds

α-tricalcium phosphate (α-TCP) paste with proper rheological characteristics for stacking a stable three-dimensional (3D) structure through a 3D printing system was formulated by mixing ground powder with 1 wt% solution of hydroxypropyl methyl cellulose prepared in 30% ethanol. The powder to liquid ratio was set to 1.67 for effective extrusion. α-TCP scaffolds were fabricated using a paste extruding deposition (PED) system, which is a representative 3D printing system, and were dried at 37 °C for 24 h before cementation. Pure calcium deficient hydroxyapatite (CDHA) scaffolds were prepared by hydrolyzing of α-TCP scaffolds in PBS solution for 24 h and then washed with deionized water several times and dried completely at room temperature for three days prior to further experiments. Schematic diagram of CDHA scaffold fabrication is shown in Appendix A.

### 3.7. Cell Imaging

MC3T3-E1 and HeLa cells were seeded and grown in 8-chambered culture slide (SPL) for 2 days to obtain adequate confluency. For differentiation of MC3T3-E1 cells, fresh differentiation medium was repeatedly supplemented every 2 or 3 days for 3 weeks. After incubation, 10 µg/mL of OB-Fn in serum-free medium were added to the cells for 1 h at 37 °C. Cells were then washed twice with DPBS (Dulbecco’s Phosphate-Buffered Saline) and fixed with 4% paraformaldehyde for 10 min, followed by washing with DPBS. The fluorescent images were acquired by EVOS^®^ FL cell imaging system (Thermo Fisher Scientific Inc., Waltham, MA, USA) using Cy5.5 light cube (Ex: 655/46, Em: 744/160).

### 3.8. Binding Assay on Hydroxyapatite

Hydroxyapatite resin and CDHA scaffold (prepared as mentioned above) were used for determining the binding of the nanoparticles to hydroxyapatite. Cy5-labeled OB-Fn, HA-Fn, or WT-Fn were added to a buffer (20 mM Tris, 50 mM NaCl, 0.05% Tween 20, pH 7.5) containing 2.5 mg/mL of hydroxyapatite resin with size less than 200 nm, followed by incubation for 1 h at room temperature. The hydroxyapatite resin was subsequently washed twice by centrifugation at 1000 rpm for 2 min for eliminating the remaining unbound ferritin nanoparticles and resuspended in the buffer. The fluorescent signal for hydroxyapatite resin was measured using fluorescence reader (SpectraMax M2e, Molecular Devices, San Jose, CA, USA) at an excitation wavelength of 600 nm and emission wavelength of 670 nm. The binding ratio of Cy5-labeled ferritin nanoparticles was defined as the fluorescent signal from hydroxyapatite resin after the reaction divided by the initial fluorescent signal. For further evaluation of binding to hydroxyapatite, each of the scaffolds was placed in a 24-well plate and immersed in a solution (20 mM Tris, 50 mM NaCl, 0.05% Tween 20, pH 7.5) containing 12.5 µg/mL of Cy5-labeled OB-Fn, HA-Fn, or WT-Fn. After incubation for 1 h at room temperature, the scaffolds were washed twice with the buffer. The fluorescent images of the scaffolds were obtained using an EVOS^®^ FL cell imaging system.

### 3.9. In Vivo Analysis

BALB/c nude mice were purchased to investigate bone-specific localization of ferritin nanoparticles in vivo. Two hundred µL of Cy5-labeled ferritin nanoparticles were injected into the mice at a dose of 10 mg/kg, while PBS was administered to the control group through intraperitoneal injection. After 24 h, mice were sacrificed and organs were then collected from each mouse for fluorescence imaging; the images were obtained using an IVIS Lumina II in vivo imaging system (Perkin Elmer, Waltham, MA, USA). All animals were cared for in accordance with the guidelines provided by the Korea Research Institute of Bioscience and Biotechnology (KRIBB), and all experiments using mice were approved by KRIBB-IACUC (approval number: KRIBB-AEC-19084).

## 4. Conclusions

In summary, we report newly engineered ferritin nanoparticles as a material for bone-targeting. The ferritin nanoparticles were genetically modified with the addition of osteoblast or hydroxyapatite binding peptides at N-terminus of ferritin subunits for this purpose. We specifically used the osteoblasts as they can directly be targeted for delivery of anabolic drugs to promote bone formation; on the contrary, hydroxyapatite was used as a target as it is the main component of the bone tissue composed of type I collagen. The results showed that the bone-targeting ferritin nanoparticles could effectively bind the respective targets as analyzed by imaging of osteoblasts and in vitro hydroxyapatite binding assay. The Cy5-labeled OB-Fn exhibited a fluorescent signal upon incubation with only the differentiated osteoblasts. Furthermore, the binding ability of HA-Fn to the hydroxyapatite resin and CDHA scaffolds was significantly higher as compared to that of OB-Fn and WT-Fn. We observed that OB-Fn and HA-Fn could serve as the promising carriers of bone-targeting systems. Ultimately, this implies that ferritin nanoparticles can be easily adapted for a specific purpose through genetic modification.

The ferritin nanoparticles may be widely applied to various biological fields, from immunoassays to medicinal sciences, due to their easy functionalization through genetic and chemical modifications. Furthermore, inorganic nanoparticles can be grown in the internal cavity of the ferritin nanoparticles to impart specific functions such as an imaging agent or a catalytic activity. Based on these results, ferritin nanoparticles can be employed not only in the diagnosis and monitoring of bone-related diseases, but also as drug delivery systems in the future.

## Figures and Tables

**Figure 1 ijms-22-04854-f001:**
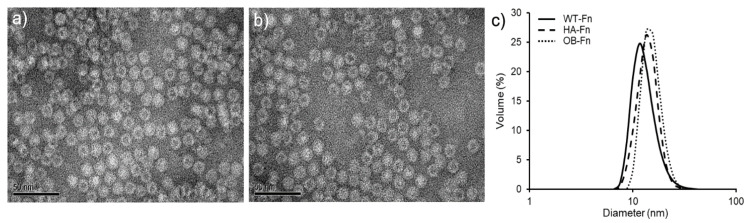
Characterization of bone-targeting ferritin nanoparticles. TEM images of (**a**) OB-Fn and (**b**) HA-Fn, (**c**) Size distribution of OB-Fn, HA-Fn, and WT-Fn measured by DLS. Scale bar 50 nm.

**Figure 2 ijms-22-04854-f002:**
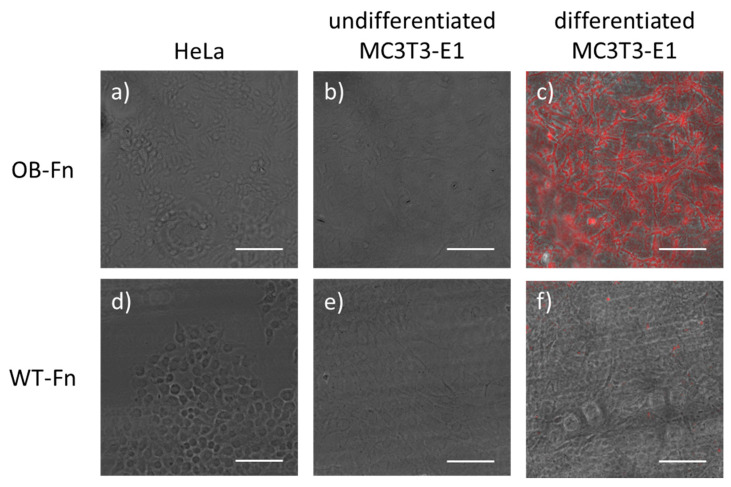
Fluorescent cell imaging for analyzing the binding specificity of OB-Fn. HeLa, undifferentiated MC3T3-E1, and MC3T3-E1 cells differentiated for 21 days were incubated with Cy5-labeled (red) (**a**–**c**) OB-Fn and (**d**–**f**) WT-Fn. Images of HeLa (**a**,**d**) and undifferentiated MC3T3-E1 (**b**,**e**) cells, are presented as merged bright field and Cy5-stained images. Images of differentiated MC3T3-E1 cells (**c**,**f**) are presented by merging only the bright field and Cy5-stained images. Scale bar 100 nm.

**Figure 3 ijms-22-04854-f003:**
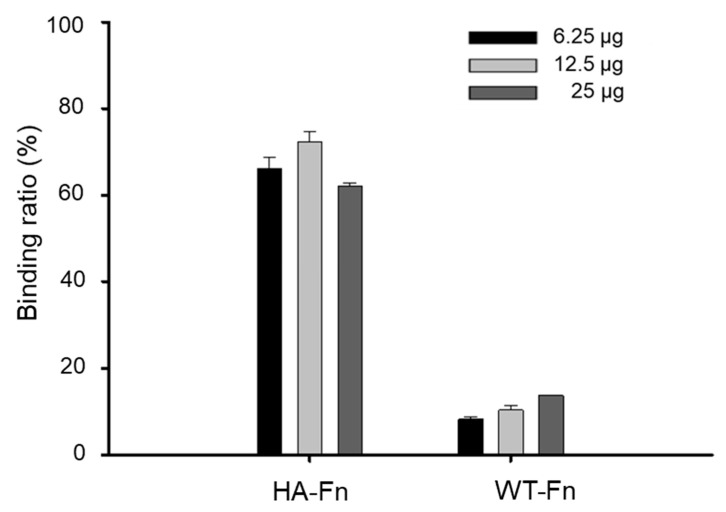
Binding of ferritin nanoparticles on hydroxyapatite resin. Cy5-labeled ferritin nanoparticles were incubated with the hydroxyapatite resins at various concentrations for 1 h. The binding ratio was defined as the fluorescent signal from resin compared with the initial fluorescent signal. Data are represented as the mean ± standard deviation (*n* = 3).

**Figure 4 ijms-22-04854-f004:**

Epifluorescent images of CDHA scaffolds. The scaffolds were immersed in the solution containing 12.5 µg/mL of Cy5-labeled (**a**) HA-Fn, (**b**) OB-Fn, and (**c**) WT-Fn. Scale bar is 500 nm.

**Figure 5 ijms-22-04854-f005:**
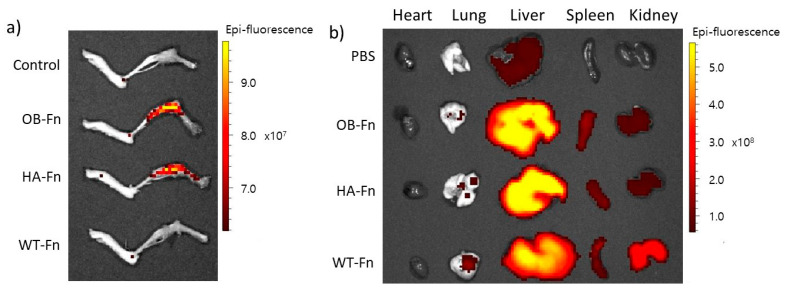
Bone-specific binding of ferritin nanoparticles. Cy5-labeled OB-Fn, HA-Fn, and WT-Fn were administered at a dose of 10 mg/kg and the images of (**a**) lower limbs and (**b**) dissected organs were acquired after 24 h. Fluorescent signals were only observed in the lower limbs of mice treated with OB-Fn and HA-Fn.

## Data Availability

Not applicable.

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
