# Peer review of "Genetically Modified Ferritin Nanoparticles with Bone-Targeting Peptides for Bone Imaging"

_ijms, 2021, doi:10.3390/ijms22094854_

Round 1

Reviewer 1 Report

The study "Genetically modified ferritin nanoparticles with bone-targeting peptides for bone imaging" is an example of a professionally and highly scholarly written study. There is a minimum of errors and shortcomings in the work and it is difficult to rebuke something. The text is written legibly, clearly and logically. Macro images lack scale (Figs. 2 and 4), please complete it. The manuscript quite noticeably lacks a suitable scheme that clearly introduces the reader to the whole concept. The number of references could have been higher.

Could the bones of the mice not be removed after the experiment and analyzed in more detail by some suitable technique (mapping) so that the fluorescence results could be independently confirmed or, if necessary, refined? Or could the results not have been refined by mapping on the lower limb cross section?

On line 244, page 7, there is no space between the number and degrees Celsius "37°C".

According to Crossref Similarity Check, the document compliance rate is 30%. This is mainly due to the similarities in Chapter 3. "Materials and Method", the Crossref Similarity Check document is attached. Authors and editors are given the opportunity to consider whether it would not be appropriate (at least in some cases) to refer to another article than to repeat the text.

Author Response

Replies to the reviewer’s comments

We are thankful for valuable comments of the reviewers, which helped us to correct errors and strengthen up our manuscript.

Reviewer #1 : The study "Genetically modified ferritin nanoparticles with bone-targeting peptides for bone imaging" is an example of a professionally and highly scholarly written study. There is a minimum of errors and shortcomings in the work and it is difficult to rebuke something. The text is written legibly, clearly and logically.

  1. Macro images lack scale (Figs. 2 and 4), please complete it.

Response : As the reviewer commented, we marked and described the scale bar in Figure. 2 and Figure. 4.

  1. The manuscript quite noticeably lacks a suitable scheme that clearly introduces the reader to the whole concept. The number of references could have been higher.

Response : To clearly introduce our whole concept to the readers, we prepared the schematic illustration as a graphical abstract instead of additional description in the introduction section.

  1. Could the bones of the mice not be removed after the experiment and analyzed in more detail by some suitable technique (mapping) so that the fluorescence results could be independently confirmed or, if necessary, refined? Or could the results not have been refined by mapping on the lower limb cross section?

Response : In this paper, we intended to demonstrate that ferritin nanoparticles were functionalized with bone targeting peptides through genetic modifications, and the resulting OB-Fn and HA-Fn could bind to bone, specifically osteoblasts and hydroxyapatite which compose bone tissue, respectively. However, fluorescent signals of dyes which have visible spectrum were hindered and rarely observed due to autofluorescence generated from skins during in vivo imaging. Therefore, to confirm bone-specific binding of ferritin nanoparticles, lower limbs and other organs were dissected from sacrificed mice and fluorescent signals from lower limbs were definitely observed.

  1. On line 244, page 7, there is no space between the number and degrees Celsius "37°C".

Response : As the reviewer commented, we inserted the space between the number and degree on line 252, page 7.

  1. According to Crossref Similarity Check, the document compliance rate is 30%. This is mainly due to the similarities in Chapter 3. "Materials and Method", the Crossref Similarity Check document is attached. Authors and editors are given the opportunity to consider whether it would not be appropriate (at least in some cases) to refer to another article than to repeat the text.

Response : To consider the presented circumstance, we performed the fabrication of CDHA scaffolds in our previous report and similarly wrote methodology in section 3.6 in this manuscript. Therefore, as the reviewer commented, we moved section 3.6 to supplementary information in Figure. S1 to avoid the repetitive description in our previous report.

Reviewer #2 : The paper is well written, organized and the methods are described in detail, being appropriate for the aim of the paper.

In my opinion, some findings from the supplementary material can be moved to the main text. Furthermore, more comments, comparison and discussion about other literature strategies, to some potential drawbacks of the authors strategy and to future applications should be provided in order to add more value to this interesting paper.

Response : As the reviewer suggested, we moved biodistribution data in Figure. S3 to the main manuscript in Figure. 5b for better demonstration. Also, we added description regarding discussion on some potential drawbacks and future applications of our strategy in the text in the revised manuscript (page 6, lines 196 ~ 204).

Reviewer 2 Report

Dear Editor,

thank you for choosing me for this rather interesting work. 

The paper is well written, organized and the methods are described in detail, being appropriate for the aim of the paper. 

In my opinion, some findings from the supplementary material can be moved to the main text. Furthermore, more comments, comparison and discussion about other literature strategies, to some potential drawbacks of the authors strategy and to future applications should be provided in order to add more value to this interesting paper. 

Best regards,

Matteo Bruno Lodi

Author Response

(The authors gave the same response as above.)
